# Learning Spectral-Decomposed Tokens for Domain Generalized Semantic Segmentation

ACM MM 2024 Anonymous Submission

## Abstract

The rapid development of Vision Foundation Model (VFM) brings inherent out-domain generalization for a variety of down-stream tasks. Among them, domain generalized semantic segmentation (DGSS) holds unique challenges as the cross-domain images share common pixel-wise content information (i.e., semantics) but vary greatly in terms of the style (e.g., urban landscape, environment dependencies). Effectively fine-tuning VFM for DGSS has recently become an open research topic for the vision community. In this paper, we present a novel **S**pectral-d**E**composed **T**oken (SET) learning framework to advance the frontier. Delving into further than existing *fine-tuning token & frozen backbone* paradigm, the proposed SET especially focuses on the way learning style-invariant features from these learnable tokens. Particularly, the frozen VFM features are first decomposed into the phase and amplitude components in the frequency space, which mainly contain the information of content and style, respectively, and then separately processed by learnable tokens for task-specific information extraction. After the decomposition, style variation primarily impacts the token-based feature enhancement within the amplitude branch. To address this issue, we further develop an attention optimization method to bridge the gap between style-affected representation and static tokens during inference. Extensive cross-domain experiments under a variety of backbones and VFMs show the state-of-the-art performance. We will make the source code publicly available.

## CCS Concepts

• **Computer Vision**; • **Representation Learning**; • **Frequency Decoupling**;

## Keywords

Domain Generalized Semantic Segmentation, Frequency Decoupling, Token Tuning, Vision Foundation Model

**ACM Reference Format:**

ACM MM 2024 Anonymous Submission. 2018. Learning Spectral-Decomposed Tokens for Domain Generalized Semantic Segmentation. In *Proceedings of Make sure to enter the correct conference title from your rights confirmation emai (Conference acronym 'XX).* ACM, New York, NY, USA, 10 pages. https://doi.org/XXXXXXX.XXXXXXX

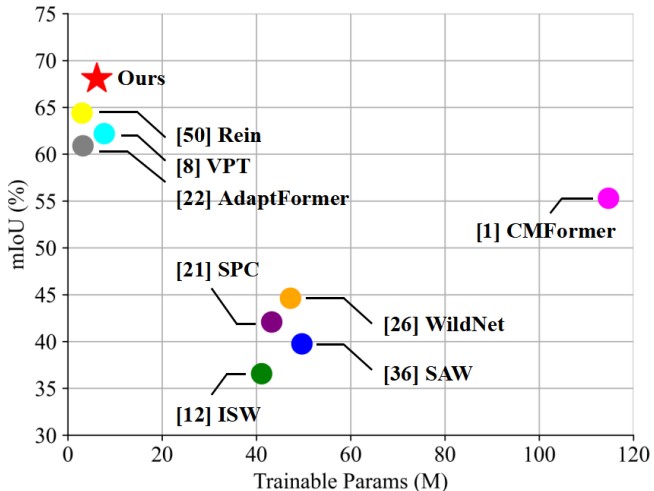

**Figure 1: Segmentation performance on unseen target domain (in mIoU, %) v.s. trainable parameter number (in million, M). GTAV [43] is used as the source domain and CityScapes [14] is used as unseen target domain. The proposed SET shows state-of-the-art performance with little trainable parameters.**

## 1 Introduction

The emergence of various Vision Foundational Models (VFMs) has started a new era for semantic segmentation. Most of these methods assume that the unseen target domains in the reference stage share the independent and identical distribution (i.i.d.) with the accessed source domain in the training stage. In the real-world applications like autonomous driving, this assumption usually does not necessarily hold. As a matter of fact, large style variations between the source and unseen target domains can be witnessed due to many shifting factors such as urban landscape, weather, and lighting conditions [10, 28, 45].

Domain generalized semantic segmentation (DGSS) is the task to address this challenge, in which the segmentation models are trained only on the source domain but inferred on arbitrary unseen target domains [1, 2, 16]. Before the VFM era, extensive efforts have been made on DGSS, which focus on either style decoupling [1, 2, 12, 16, 36, 37] or style augmentation [21, 26, 60, 62]. However, the representation ability of the scene content (e.g., semantics) itself, which is relatively stable between different domains, remains less explored. In recent years, VFMs (e.g., CLIP [41], DALL-E [42], DINOv2 [32], and SAM [24]) have significantly advanced a variety of vision tasks. The inherited generalization ability of VFM from large-scale image pre-training [25] has great potential to be harnessed for DGSS.

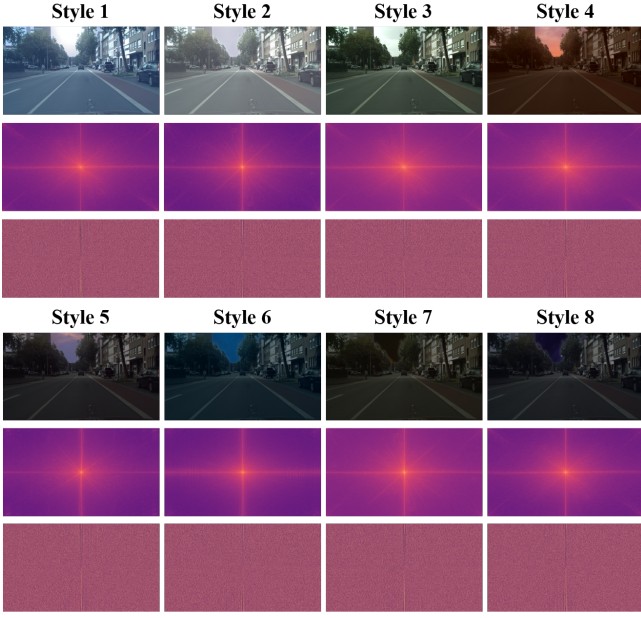

**Figure 2: A toy example on how different styles of a driving scene (left row) impact the amplitude (second row) and the phase component (third row). As the styles change, the phase component remains stable, while the amplitude adjusts correspondingly. Thus, the amplitude component provides a feasible path to inspect the cross-domain style variation.**

Recent works show that, in the context of DGSS, fine-tuning VFM with learnable tokens yields better generalization than the one with full parameter tuning [51]. However, the key challenge of DGSS, i.e., the domain gap caused by the style variation, remains unaddressed. Hence, we naturally raise an open research question: *how to learn style-invariant representation by fine-tuning VFMs?* Spectral decomposition has been long acknowledged effective to handle the style and content information separately in the context of domain generalization [53, 55], where the style/content information highly rests in the high-/low- frequency component [49]. This further leads to our second research question: *how to design a frequency-space-based method to effectively composite the style and content information from the VFM features?*

In this paper, we propose a novel **S**pectral-d**E**composited **T**oken (SET) learning scheme to address this challenge. Fast fourier transform (FFT) [3] is implemented to decompose the image feature from the previous frozen VFM layer to the amplitude and phase components. The phase component, which contains more low-frequency component on the scene content, is relatively stable despite the cross-style variation [38, 54]. In contrast, the amplitude component, which has more high-frequency component on the styles [6, 55], can be rather unstable when handling images from different domains. Therefore, the token-based fine-tuning in the amplitude branch deserves more attentions in domain generalization.

To this end, we design a token-based feature enhancement pipeline which extracts task-specific information and adjusts the spectral-decomposed features in amplitude and phase branches separately.

Learnable tokens capture task-relevant features within each branch, and enhance the original feature representations based on their similarities. During inference, these tokens are fixed. However, since the amplitude component is susceptible to style variation, the enhancement process in the amplitude branch can become unstable due to the fluctuating representation and static tokens. Therefore, we further propose an attention optimization method to make the token-based enhancement more robust to style changes. Finally, the enhanced amplitude and phase components are projected back to the spatial space and input to the subsequent layer.

We conduct extensive experiments in various domain-generalized semantic segmentation scenarios, where models are trained on one dataset from [14, 30, 43, 44, 56] as the source domain and validated on the remaining four datasets as unseen target domains. Although all datasets share the same 19 semantic categories, they differ in scene styles. Results demonstrate that our proposed SET achieves up to an improvement of: 1) 20% on mIoU over state-of-the-art CNN-based methods [26, 36]; 2) 10% mIoU over Mask2Former based methods [1, 16]; 3) up to 3.12% mIoU over the VFM based Rein [51] on unseen ACDC-snow domain. Additionally, the proposed SET can be seamlessly transferred to a variety of VFMs with good generalization capability.

Our contributions are summarized as follows:

- We propose a novel spectral-decomposed token learning (SET) scheme for DGSS. It harnesses the style-invariant properties for VFM.
- The proposed SET, consisting of three key steps, namely, spectral decomposition, learning spectral tokens, and attention optimization in the amplitude branch, can be seamlessly integrated into existing VFMs in a trainable manner.
- Extensive experiments show the proposed SET outperforms the VFM based state-of-the-art by up to 1.66% and 3.12% mIoU on unseen CityScapes and ACDC-snow domains.

## 2 Related Work

### 2.1 Domain Generalization

Domain generalization focused on the scenarios where the target domain is unavailable during training. Various approaches have been proposed in the past decade. A straight-forward research line is to design variations of the normalization techniques [13, 29, 46], which are simple and effective to enhance the representation robustness to style robustness. Additionally, extensive advanced techniques, to name a few, adversarial training [15], domain alignment [9, 47], meta-learning [7, 40], data augmentation [48, 58, 61], self-supervised learning [4], and regularization techniques [23, 50], have been adapted to learn the domain-invariant representation.

However, the above works usually focus on the non task-specific settings, not especially devised for DGSS. In DGSS, the key challenge lies in that the cross-domain images share common content information (i.e., semantics) but vary greatly in terms of the style variation (e.g., urban landscape, environment dependencies).

### 2.2 Vision Foundation Models

Foundation model is a new paradigm for deep learning. Its key idea is to pre-train a deep network on a board set of unlabeled

images, which has a strong representation ability to be fine-tuned on a variety of down-stream tasks. This paradigm firstly emerges in the field of Natural Language Processing (NLP), and later also draws increasingly attention in the community of computer vision. For simplicity, in the following text, the foundation models in the computer vision field is termed as Vision Foundation Model (VFM).

Here, we review several typical VFMs in the past few years: Contrastive Language-Image Pre-Training (CLIP) [41] acquires high-fidelity visual representations via contrastive learning with large-scale image-text pairs. Masked Auto-encoder (MAE) [18] employs a masked image modeling framework to derive latent image representations. Segment Anything Model (SAM) [24] pioneers a promptable model pre-trained on a diverse dataset for segmentation tasks. Explore the limits of Visual representation at scAle (EVA) [17] merges Masked Image Modeling pre-training with CLIP's vision features as the target of pretext tasks. Self-DIstillation with NO labels (DINO) [32] pre-trains the deep model on extensive, meticulously curated datasets without explicit supervision.

To summarize, these VFMs have shown great successes on improving a variety of downstream tasks, underscoring their remarkable generalization capabilities. Nevertheless, an in-depth exploration of their effectiveness in the specialized realm of DGSS tasks remains rarely explored. The recent-developed Rein shows that fine-tuning VFM [8, 22] with learnable tokens yields the better generalization than the one with full parameter tuning [51].

## 2.3 Domain Generalized Semantic Segmentation

Semantic segmentation in driving scenes can encounter great domain shift, caused by factors such as adverse weather, diverse illumination, and urban landscape diversity. In the past decade, domain adaptation methods [39, 59] have shown great successes, but they can only generalize to the target domain that has learned in the training stage. To generalize to arbitrary unseen target domains, DGSS has drawn increasing attentions in the past few years.

In the convolutional neural network (CNN) era, DGSS methods can be categorized into two types, namely style decoupling and style augmentation. For style-decoupling-based methods, instance normalization (e.g., IBN [33], Iternorm [20]) and instance whitening (e.g., IW [34], SAW [36], ISW [12], DIRL [52]) operations are commonly used. For style-augmentation-based methods, external training images from other sources are usually used to enrich the domain diversity [21, 26, 37, 57, 60, 62]. Later in the Vision Transformer (ViT) era, DGSS methods usually leverage the strong content representation ability of mask attention mechanism [1, 2, 16], which learns a more global-wise representation than convolution and is more robust to the cross-domain style variation.

Nevertheless, the exploration of DGSS methods based on VFMs remains relatively limited. A recent work Rein shows that, fine-tuning VFM with learnable tokens yields better generalization for DGSS than full parameter tuning [51].

## 3 Methodology

### 3.1 Problem Definition

In DGSS, there are source domain $S = (x_s, y_s)$ and a number of target domains $U_1, U_2, \cdots, U_n$, which have rather different feature distributions. Here $x_s$ represents an image and $y_s$ denotes its corresponding pixel-wise label. The target domain $U$ is inaccessible during training. Our objective is to train a VFM-based semantic segmentation model which is capable of achieving superior generalization performance when inferring on these target domains.

The overview of the proposed **S**pectral-d**E**composed **T**okens (SET) framework is shown in Fig. 3. To efficiently fine-tune VFMs, spectral-decomposed tokens are applied to each frozen layer. Within each layer, the frozen features are decomposed into amplitude and phase components by Fast Fourier Transform (FFT) (in Sec. 3.2). Spectral tokens are linked to amplitude and phase branches to extract the task-specific information and enhance the original features (in Sec. 3.3). Additionally, since the style variation mainly affects the amplitude component, we further design an attention optimization method to improve the generalization ability of the learned tokens (in Sec. 3.4). Finally, two enhanced components are projected back to the spatial space by Inverse Fast Fourier Transform (IFFT) and fed into the subsequent layer.

### 3.2 Spectral Decomposition

As discussed above, the low-frequency and high-frequency components in the frequency space provide a feasible solution to separate the content and style information. Inspired by this, we turn to the spectral decomposition [5, 35] via Fast Fourier Transform (FFT) to realize this objective. Specifically, given the intermediate image features $X_k \in \mathbb{R}^{d \times H \times W}$ of layer $k$, it is fed to a 2D Fast Fourier Transform, independently to each channel to obtain the corresponding projected representations $X_f \in \mathbb{R}^{H \times W}$ in frequency space. This process on $X_i$ within each channel can be computed as

$$X_f(x, y) = \sum_{h=0}^{H-1} \sum_{w=0}^{W-1} X_i(h, w) e^{-2j\pi(x\frac{h}{H} + y\frac{w}{W})}, \quad (1)$$

where $H$ and $W$ are the height and the width of the image features. In addition, $d$ denotes the feature dimension.

On the other hand, the inverse transformation from the frequency space to the spatial space is usually implemented by the Inverse Fast Fourier Transform (IFFT), which is mathematically computed as

$$X_k(h, w) = \frac{1}{HW} \sum_{x=0}^{H-1} \sum_{y=0}^{W-1} X_f(x, y) e^{2j\pi(x\frac{h}{H} + y\frac{w}{W})}. \quad (2)$$

The frequency representation $X_f$ can be divided into two parts, namely the real part $X_f^{real}$ and the imagery part $X_f^{img}$ respectively, defined as

$$X_f(x, y) = X_f^{real} + iX_f^{img}. \quad (3)$$

Decomposing the image feature $X_i$ into its amplitude $\alpha$ and phase $\rho$ is known as spectral decomposition, given by

$$\alpha = \sqrt{(X_f^{real})^2 + (X_f^{img})^2},$$
$$\rho = \arctan(X_f^{img}/X_f^{real}). \quad (4)$$

Through this series of mathematical transformations, the frequency representation $X_f$ can be presented as

$$X_f = \alpha cos(\rho) + i\alpha sin(\rho). \quad (5)$$

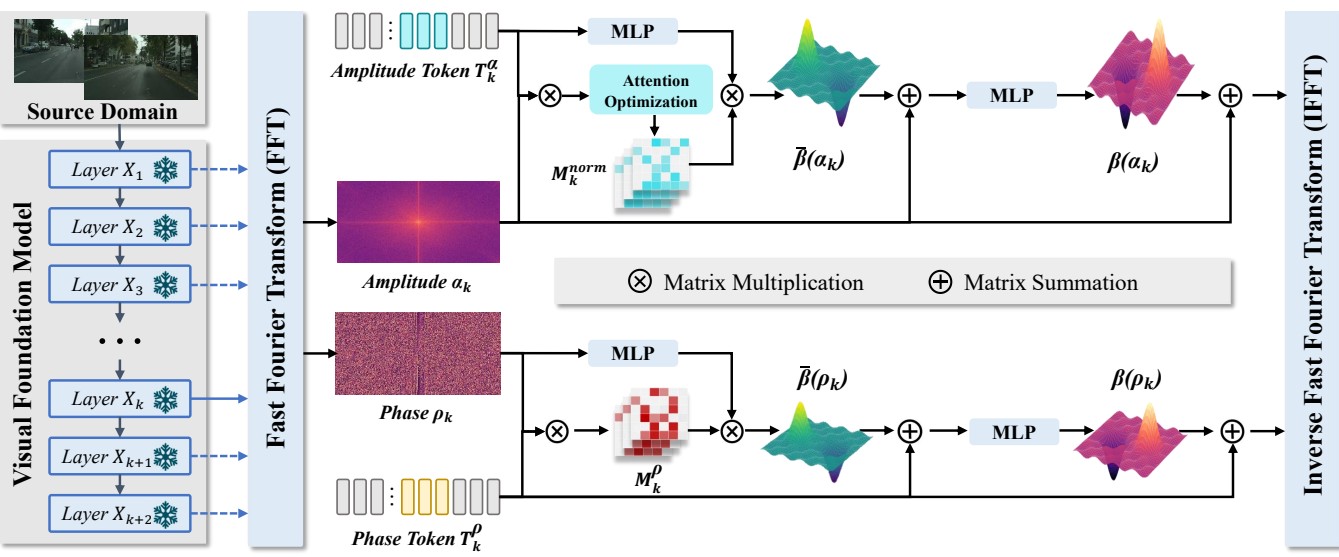

**Figure 3: Framework overview of the proposed Spectral-dEcomposed Token (SET) learning scheme. Embedded into each frozen layer of a VFM, the proposed SET consists of three steps, namely spectral decomposition (in Sec. 3.2), learning spectral tokens (in Sec. 3.3) and attention optimization in amplitude branch (in Sec. 3.4). By default, DINOv2 [32] is used as the frozen VFM, while the proposed SET is versatile to different VFMs.**

Define the function to decompose the frequency feature into its amplitude and phase components as $decompose(\cdot)$, and the opposite process as $compose(\cdot)$, this process can be presented as

$$\alpha, \rho = decompose(X_k),$$
$$X_k = compose(\alpha, \rho). \tag{6}$$

### 3.3 Learning Spectral Tokens

Leveraging learnable tokens with only a few number of parameters [19] has been turned out to be an effective path to fine-tune the VFMs. Following this simple yet effective paradigm, in our pipeline, learnable tokens are utilized to refine the spectral components $\alpha_k$ and $\rho_k$ of features at each layer within the frozen VFM backbone. Since these tokens refine the original features in the frequency domain, we name them as spectral tokens in this paper. Specifically, for the features $X_k$ generated by the $k$-th layer $V_k$, spectral components $\alpha_k, \rho_k$ is obtained by spectral decomposition. The enhanced spectral features $\hat{\alpha}_k, \hat{\rho}_k$ are produced by token-based adjustment $\beta(\cdot)$ and composed to output feature as the input for the subsequent layer, given by

$$\alpha_k, \rho_k = decompose(X_k),$$
$$\hat{\alpha}_k = \alpha_k + \beta(\alpha_k), \hat{\rho}_k = \rho_k + \beta(\rho_k), \tag{7}$$
$$X_{k+1} = V_{k+1}(compose(\hat{\alpha}_k, \hat{\rho}_k)).$$

Assume we have a set of spectral tokens $T = \{T_i^D \in \mathbb{R}^{l \times d} | D \in [\alpha, \rho], 1 \le i \le N\}$, where $T_k^\alpha$ and $T_k^\rho$ denote the learnable tokens for $\alpha_k$ and $\rho_k$, respectively. $N$ is the number of layers in VFM, $l$ denote the length of each token (number of features in each token), and $d$ denote the feature dimension of $T_i$, which is equal to the channel number in spectral-decomposed features. During training, the backbone is kept frozen and the task-specific posterior

is acquired from the DGSS dataset through these spectral tokens, bridging the scene disparity concerning the pre-training datasets and fine-tuning datasets.

To achieve this, each spectral token includes a bag of learnable features to capture the task-specific knowledge in frequency domain. Within the token-based adjustment process $\beta(\cdot)$, these bag features are used to enhance the spectral-decomposed features based on their similarity. Through this enhancement, task-relevant representations in the original features are further highlighted, while task-irrelevant category information is partially filtered out.

Specifically, inner-product is utilized to measure the similarity between token features and original features from frozen layers. Similarity map $M_k^\alpha \in \mathbb{R}^{HW \times l}$ is built to capture the association between spectral-decomposed features ($\alpha_k$ as example) and spectral token (amplitude token $T_k^\alpha$ as example). Following the widely-used attention mechanism, a softmax function is applied to each line of $M_k^\alpha$ to normalize the weights of token features. To summarize, the feature-token similarity can be mathematically computed as

$$M_k^\alpha = Softmax(\frac{\alpha_k \times T_k^\alpha}{\sqrt{d}}). \tag{8}$$

Through the feature-token similarity map $M_k^\alpha$, we can select the relevant token features for each position of the spectral-decomposed features. These relevant token features include learned task-specific knowledge which is used to enhance the original features. Before added into the frozen spectral-decomposed features, the token features are further processed by a Multi-Layer Perceptron (MLP) layer to generate a more suitable representation for enhancement. This intermediate process $\overline{\beta}(\alpha_k)$ can be presented as

$$\overline{\beta}(\alpha_k) = M_k^\alpha \times MLP(T_k^\alpha). \tag{9}$$

Finally, another MLP layer is adapted to the enhanced features. This layer is designed to extract task-related information and filter out the irrelevant representations. The overall process of token-based adjustment $\beta(\cdot)$ can be written as

$$\beta(\alpha_k) = MLP(\alpha_k + \overline{\beta}(\alpha_k)),$$
$$\beta(\rho_k) = MLP(\rho_k + \overline{\beta}(\rho_k)). \tag{10}$$

## 3.4 Attention Optimization

During inference, the parameters of the learnable tokens are fixed. As shown in Fig. 2, the style variations can significantly reflect in the amplitude component. As a result, when dealing with images from unseen target domains, the weights of the feature-token similarity map in the amplitude branch may be severely affected by the style change. Since the task-related category information is stored in the tokens features, the perturbation of weights leads to the injection of incorrect category information during the enhancement process, thereby weakening the domain generalization ability. To address this problem, we further proposed an attention optimization method to adjust the similarity map in amplitude branch.

Specifically, we perform a further normalization on the feature-token similarity map,

$$M_k^{norm} = \frac{M_k^\alpha - \mu}{\sigma}, \tag{11}$$

where $\mu$ ans $\sigma$ denote the mean and standard deviation of $M_k^\alpha$. Let $H, W$ denote the size of similarity map, $\mu$ ans $\sigma$ are computed by

$$\mu = \frac{1}{HW} \sum_{i=1}^{H} \sum_{j=1}^{W} M_{i,j},$$
$$\sigma^2 = \frac{1}{HW} \sum_{i=1}^{H} \sum_{j=1}^{W} (M_{i,j} - \mu)^2. \tag{12}$$

Due to the softmax operation when acquiring $M_k^\alpha$, the original mean value of $M_k^\alpha$ is $\frac{1}{l}$ and the max standard deviation is $\sqrt{\frac{1}{l}(1 - \frac{1}{l})}$, where $l$ is the number of features in each token. After normalization, the standard deviation increases to 1, which means higher weights are assigned to the relevant token features and the weight distribution becomes more uneven. When amplitude feature representation affected by style variation, its similarity with fixed tokens trained on source dataset decreases, leading to more uniform weight distribution within the feature-token similarity map. By the proposed attention optimization, the weight distribution is refined and the relevant token features receive more attention, alleviating the impacts of style change for token-based fine-tuning. After that, the enhancement process in amplitude branch can be written as

$$\hat{\alpha}_k^{norm} = \alpha_k + MLP(\alpha_k + M_k^{norm} \times MLP(T_k^\alpha)). \tag{13}$$

Finally, amplitude component is combined with phase component, projected back to the spatial space and fed into the next layer in VFM, given by

$$X_{k+1} = V_{k+1}(compose(\hat{\alpha}_k^{norm}, \hat{\rho}_k)). \tag{14}$$

## 3.5 Implementation Details

Following prior work [51], the model is trained 40000 iterations with a batch size of 4 and an initial learning rate of 1e-4 for DGSS tasks. The resolution of input images is 512×512. DINOv2 is chosen as the default VFM, and the segmentation head of Mask2Former [11] is utilized to produce pixel-level prediction.

## 4 Experiment

### 4.1 Datasets & Evaluation Protocols

*4.1.1 Datasets* We conduct the experiments on five driving-scene semantic segmentation datasets.

**CityScapes** [14] is constructed based on the driving-scenes in Germany cities, which includes 2,975 and 500 well-annotated samples for training and validation, respectively. The resolution of CityScapes is 2,048×1,024.

**BDD-100K** [56] provides diverse scenes of driving videos under various weather conditions. It contains 7,000 and 1,000 fine-annotated samples for training and validation of semantic segmentation, respectively. The resolution of BDD-100K is of 1,280×720.

**SYNTHIA** [44] provides a large-scale synthetic dataset, and provides 9,400 images with a resolution of 1,280×760.

**Mapillary** [30] provides a large-scale semantic segmentation dataset based on street scenes with 25,000 samples.

**GTAV** [43] is a synthetic semantic segmentation dataset rendered by the GTAV game engine. It provides 24,966 simulated urban-street samples with a resolution of 1,914×1,052.

*4.1.2 Evaluation Settings* We illustrate our domain generalization settings as follows. Firstly, we use C, B, S, M and G to denote the above five datasets respectively. Following prior DGSS works [12, 33, 34, 36], the segmentation model is trained on one dataset as the **source domain**, and is validated on the rest of the four datasets as the **target domains**. Three settings include: 1) G → {C, B, M, S}; 2) S → {C, B, M, G}; and 3) C to → {B, M, G, S}. We employ the mIoU (%) metric for the evaluation. All the reported performance is directly cited from prior works [12, 33, 34, 36].

*4.1.3 Baselines* Existing DGSS methods are included for comparison, namely, IBN [33], IW [34], Iternorm [20], DRPC [57], ISW [12], GTR [37], DIRL [52], SHADE [60], SAW [36], WildNet [26], AdvStyle [62], SPC [21], HGFormer [16], CMFormer [1], DIDEX [31] and Rein [51].

## 4.2 Comparison with State-of-the-art

*4.2.1 GTAV Source Domain* Table 1 compares the performance of the proposed SET with existing state-of-the-art DGSS methods under the G → {C, B, M, S} setting. The proposed SET shows an mIoU improvement of 1.66%, 1.24%, 1.58% and 1.15% on C, B, M and S unseen target domains, respectively, compared to the VFM based state-of-the-art Rein [51]. In addition, the mIoU improvements on ResNet based and Mask2Former based DGSS methods are more than 20% and 10%, respectively. It is worthy noting that the source domain GTAV is a synthetic dataset, while C, B and M target domains are real datasets. The positive outcomes under this setting demonstrates the feature generalization ability of the proposed SET.

| Method | Venue | Trained on GTAV (G) | | | |
|---|---|---|---|---|---|
| | | → C | → B | → M | → S |
| *ResNet based:* | | | | | |
| IBN [33] | ECCV 2018 | 33.85 | 32.30 | 37.75 | 27.90 |
| IW [34] | CVPR 2019 | 29.91 | 27.48 | 29.71 | 27.61 |
| Iternorm [20] | CVPR 2019 | 31.81 | 32.70 | 33.88 | 27.07 |
| DRPC [57] | ICCV 2019 | 37.42 | 32.14 | 34.12 | 28.06 |
| ISW [12] | CVPR 2021 | 36.58 | 35.20 | 40.33 | 28.30 |
| GTR [37] | TIP 2021 | 37.53 | 33.75 | 34.52 | 28.17 |
| DIRL [52] | AAAI 2022 | 41.04 | 39.15 | 41.60 | - |
| SHADE [60] | ECCV 2022 | 44.65 | 39.28 | 43.34 | - |
| SAW [36] | CVPR 2022 | 39.75 | 37.34 | 41.86 | 30.79 |
| WildNet [26] | CVPR 2022 | 44.62 | 38.42 | 46.09 | 31.34 |
| AdvStyle [62] | NeurIPS 2022 | 39.62 | 35.54 | 37.00 | - |
| SPC [21] | CVPR 2023 | 44.10 | 40.46 | 45.51 | - |
| *Mask2Former based:* | | | | | |
| CMFormer [1] | AAAI 2024 | 55.31 | 49.91 | 60.09 | 43.80 |
| *VFM based:* | | | | | |
| DIDEX*[31] | WACV 2024 | 62.0 | 54.3 | 63.0 | - |
| REIN*[51] | CVPR 2024 | 66.4 | 60.4 | 66.1 | 48.86† |
| **Ours** | - | **68.06** | **61.64** | **67.68** | **50.01** |
| | | ↑1.66 | ↑1.24 | ↑1.58 | ↑1.15 |

**Table 1: G → {C, B, M, S} setting. Performance comparison between the proposed SET (ours) and existing DGSS methods. '-': The metric is either not reported or the official source code is not available. Evaluation metric mIoU is given in (%). '*': only one decimal result is reported. '†': results are re-implemented.**

| Method | Venue | Trained on SYNTHIA (S) | | | |
|---|---|---|---|---|---|
| | | → C | → B | → M | → G |
| *ResNet based:* | | | | | |
| IBN [33] | ECCV 2018 | 32.04 | 30.57 | 32.16 | 26.90 |
| IW [34] | CVPR 2019 | 28.16 | 27.12 | 26.31 | 26.51 |
| DRPC [57] | ICCV 2019 | 35.65 | 31.53 | 32.74 | 28.75 |
| ISW [12] | CVPR 2021 | 35.83 | 31.62 | 30.84 | 27.68 |
| GTR [37] | TIP 2021 | 36.84 | 32.02 | 32.89 | 28.02 |
| SAW [36] | CVPR 2022 | 38.92 | 35.24 | 34.52 | 29.16 |
| AdvStyle [62] | NeurIPS 2022 | 37.59 | 27.45 | 31.76 | - |
| *Mask2Former based:* | | | | | |
| CMFormer [1] | AAAI 2024 | 44.59 | 33.44 | 43.25 | 40.65 |
| *VFM based:* | | | | | |
| REIN†[51] | CVPR 2024 | 48.59 | 44.42 | 48.64 | 46.97 |
| **Ours** | - | **49.65** | **45.45** | **49.45** | **48.05** |
| | | ↑1.06 | ↑1.03 | ↑0.81 | ↑1.08 |

**Table 2: S → {C, B, M, G} setting. Performance comparison between the proposed SET (ours) and existing DGSS methods. '-': The metric is either not reported or the official source code is not available. Evaluation metric mIoU is given in (%). '*': only one decimal result is reported. '†': results are re-implemented.**

*4.2.2 SYNTHIA Source Domain* Table 2 compares the performance of the proposed SET and existing state-of-the-art DGSS methods under the S → {C, B, M, G} setting. The proposed SET shows improvements of 1.06%, 1.03%, 0.81% and 1.08% on mIoU against the runner-up method Rein [51]. In addition, the mIoU improvements

| Method | Venue | Trained on Cityscapes (C) | | | |
|---|---|---|---|---|---|
| | | → B | → M | → G | → S |
| *ResNet based:* | | | | | |
| IBN [33] | ECCV 2018 | 48.56 | 57.04 | 45.06 | 26.14 |
| IW [34] | CVPR 2019 | 48.49 | 55.82 | 44.87 | 26.10 |
| Iternorm [20] | CVPR 2019 | 49.23 | 56.26 | 45.73 | 25.98 |
| DRPC [57] | ICCV 2019 | 49.86 | 56.34 | 45.62 | 26.58 |
| ISW [12] | CVPR 2021 | 50.73 | 58.64 | 45.00 | 26.20 |
| GTR [37] | TIP 2021 | 50.75 | 57.16 | 45.79 | 26.47 |
| DIRL [52] | AAAI 2022 | 51.80 | - | 46.52 | 26.50 |
| SHADE [60] | ECCV 2022 | 50.95 | 60.67 | 48.61 | 27.62 |
| SAW [36] | CVPR 2022 | 52.95 | 59.81 | 47.28 | 28.32 |
| WildNet [26] | CVPR 2022 | 50.94 | 58.79 | 47.01 | 27.95 |
| *Mask2Former based:* | | | | | |
| HGFormer*[16] | CVPR 2023 | 53.4 | 66.9 | 51.3 | 33.6 |
| CMFormer [1] | AAAI 2024 | 59.27 | 71.10 | 58.11 | 40.43 |
| *VFM based:* | | | | | |
| REIN†[51] | CVPR 2024 | 63.54 | 74.03 | 62.41 | 48.56 |
| Ours | - | **65.07** | **75.67** | **63.80** | **49.61** |
| | | ↑1.53 | ↑1.64 | ↑1.39 | ↑1.05 |

**Table 3: C → {B, M, G, S} setting. Performance comparison between the proposed SET (ours) and existing DGSS methods. '-': the metric is either not reported or the official source code is not available. Evaluation metric mIoU is given in (%). '†': results are re-implemented.**

| Component | | | | Trained on GTAV (G) | | | |
|---|---|---|---|---|---|---|---|
| VFM | Spe. | Token | AO | → C | → B | → M | → S |
| ✓ | | | | 63.30 | 56.10 | 63.90 | 46.50 |
| ✓ | ✓ | | | 65.23 | 59.34 | 64.32 | 47.10 |
| ✓ | | ✓ | | 66.40 | 60.40 | 66.10 | 48.86 |
| ✓ | ✓ | ✓ | | 66.78 | 60.59 | 66.08 | 48.92 |
| ✓ | ✓ | ✓ | ✓ | **68.06** | **61.64** | **67.68** | **50.01** |

**Table 4: Ablation studies on key components of SET under the G → {C, B, M, S} setting. *VFM*: use only frozen VFM to predict, *Spe.*: spectral decomposition, *Token*: spectral tokens, *AO*: attention optimization.**

on ResNet based and Mask2Former based DGSS methods are more than 20% and 10%, respectively, when generalized to B, M and G unseen target domains.

*4.2.3 CityScapes Source Domain* Table 3 compares the performance of the proposed SET and existing state-of-the-art DGSS methods under the C → {B, M, G, S} setting. The proposed SET also shows a clear performance improvement than the second-best Rein [51]. Specifically, the mIoU improvements on the B, M, G and S unseen target domains are 1.53%, 1.64%, 1.39% and 1.05%, respectively. In addition, the mIoU improvements on ResNet based and Mask2Former based DGSS methods are more than 15% and 8% in average. To better understand how the proposed SET improves the feature generalization when compared with Rein [51], Fig. 4 visualizes the feature space of the Rein baseline (left) and the proposed SET (right). The proposed SET allows the samples from different unseen target domains to be more uniformly distributed.

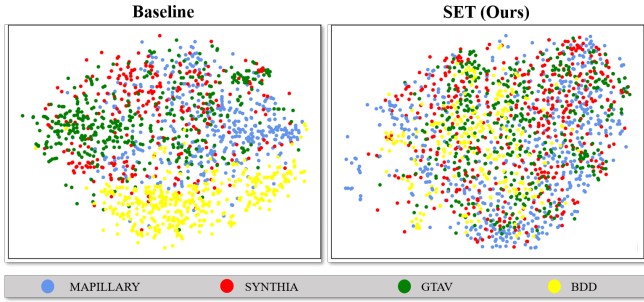

**Baseline**      **SET (Ours)**

● MAPILLARY     ● SYNTHIA     ● GTAV     ● BDD

**Figure 4: T-SNE visualization of the feature space from baseline (left), and the proposed SET (RDWT; right). The model is trained on the CityScapes source domain, and inferred on the rest four unseen target domains. The proposed SET allows the unseen target domain samples to be more uniformly distributed. Better zoom in to view.**

| Attention Optimization | | | Trained on GTAV (G) | | | |
|---|---|---|---|---|---|---|
| Image | Phase | Amplitude | → C | → B | → M | → S |
| ✓ | | | 64.91 | 56.97 | 64.20 | 46.72 |
| | ✓ | | 63.27 | 56.38 | 62.71 | 46.08 |
| | ✓ | ✓ | 66.01 | 58.83 | 64.62 | 48.16 |
| | | ✓ | **68.06** | **61.64** | **67.68** | **50.01** |

**Table 5: Ablation studies on attention optimization within different components.** *Image* **denotes the original image features,** *Phase* **denotes phase component,** *Amplitude* **denotes amplitude component.**

## 4.3 Ablation Studies

*4.3.1 On Key Components* Table 4 studies the impact of each component in the proposed SET. The experiments are conducted under the G → {C, B, M, S} setting. When only the frozen VFM (DINOv2) is used for DGSS (in the first row), the mIoU is relatively low. The spectral decomposition operation (in the second row) contributes to 1.93%, 3.24%, 0.42% and 0.60% mIou gain on the C, B, M and S target domains by separately learning the amplitude and phase information. Furthermore, when spectral tokens are implemented (in the third row), the performance gain are 3.10%, 4.30%, 3.10% and 2.36% mIou on the C, B, M and S target domains, respectively. Finally, based on the spectral tokens, attention optimization contributes to an additional improvement of 1.28%, 1.05%, 1.60% and 1.09%.

*4.3.2 On Attention Optimization* Table 5 further studies the attention optimization operation on each component. The outcomes show that, attention optimization in phase branch can potentially harm the model's representational capability. In contrast, the optimal choice is to only used for amplitude component, which shows the most predominate improvement on the generalization ability.

## 4.4 Generalization Ability Test

*4.4.1 On Different VFMs* We validate if the proposed SET can be generalized to a variety of VFMs. It is integrated into CLIP [41],

| Backbone | Fine-tune Method | Trainable Params* | mIoU | | | |
|---|---|---|---|---|---|---|
| | | | Citys | BDD | Map | Avg. |
| CLIP [41] | Full | 304.15M | 51.3 | 47.6 | 54.3 | 51.1 |
| | Freeze | 0.00M | 53.7 | 48.7 | 55.0 | 52.4 |
| | Rein [51] | 2.99M | 57.1 | 54.7 | 60.5 | 57.4 |
| | SET | 6.13M | **58.2** | **55.3** | **61.4** | **58.3** |
| MAE [18] | Full | 330.94M | 53.7 | **50.8** | 58.1 | 54.2 |
| | Freeze | 0.00M | 43.3 | 37.8 | 48.0 | 43.0 |
| | Rein [51] | 2.99M | 55.0 | 49.3 | 58.6 | 54.3 |
| | SET | 6.13M | **56.2** | 51.0 | **60.2** | **55.8** |
| SAM [24] | Full | 632.18M | 57.6 | 51.7 | 61.5 | 56.9 |
| | Freeze | 0.00M | 57.0 | 47.1 | 58.4 | 54.2 |
| | Rein [51] | 4.51M | 59.6 | 52.0 | 62.1 | 57.9 |
| | SET | 6.13M | **60.7** | **52.8** | **63.2** | **58.9** |
| EVA02 [17] | Full | 304.24M | 62.1 | 56.2 | 64.6 | 60.9 |
| | Freeze | 0.00M | 56.5 | 53.6 | 58.6 | 56.2 |
| | Rein [51] | 2.99M | 65.3 | 60.5 | 64.9 | 63.6 |
| | SET | 6.13M | **66.4** | **61.8** | **65.6** | **64.6** |
| DINOV2 [32] | Full | 304.20M | 63.7 | 57.4 | 64.2 | 61.7 |
| | Freeze | 0.00M | 63.3 | 56.1 | 63.9 | 61.1 |
| | Rein [51] | 6.13M | 66.4 | 60.4 | 66.1 | 64.3 |
| | SET | 6.13M | **68.0** | **61.6** | **67.6** | **65.7** |

**Table 6: Generalization ability test of the proposed SET on different VFM models. One decimal result is reported and compared following prior references.**

| Method | Venue | Trained on Cityscapes (C) | | | |
|---|---|---|---|---|---|
| | | → Fog | → Night | → Rain | → Snow |
| *ResNet based:* | | | | | |
| IBN* [33] | ECCV 2018 | 63.8 | 21.2 | 50.4 | 49.6 |
| IW* [34] | CVPR 2019 | 62.4 | 21.8 | 52.4 | 47.6 |
| ISW* [12] | CVPR 2021 | 64.3 | 24.3 | 56.0 | 49.8 |
| *Mask2Former based:* | | | | | |
| ISSA* [27] | WACV 2023 | 67.5 | 33.2 | 55.9 | 53.2 |
| CMFormer* [1] | AAAI 2024 | 77.8 | 33.7 | 67.6 | 64.3 |
| *VFM based:* | | | | | |
| Rein† [51] | CVPR 2024 | 79.48 | 55.92 | 72.45 | 70.57 |
| **Ours** | | **80.06** | **57.29** | **74.80** | **73.69** |
| | | ↑0.58 | ↑1.37 | ↑2.35 | ↑3.12 |

**Table 7: Generalization of the proposed CMFormer to the adverse condition domains (rain, fog, night and snow) on ACDC dataset [45]. '*': Only reports one decimal results. '†': Reports re-implementation result.**

MAE [18], SAM [24], EVA02 [17] and DINOV2 [32] under full-training, full-freezing and fine-tuning scheme, respectively. One decimal result is reported and compared following prior references. Besides, Rein [51], as the fine-tuning baseline, is also involved for comparison. Table 6 reports the outcomes under the G → {C, B, M} setting. Our SET shows a significant performance improvement than Rein [51] and other methods with all listed VFMs.

*4.4.2 On Adverse Conditions* We further inspect the generalization ability of the proposed SET to a variety of adverse conditions. Adverse Conditions Dataset with Correspondences (ACDC) (ACDC) [45] is the largest semantic segmentation dataset under adverse conditions. Following prior works, fog, night, rain and snow are set as four different unseen domains. CityScapes is used as the source

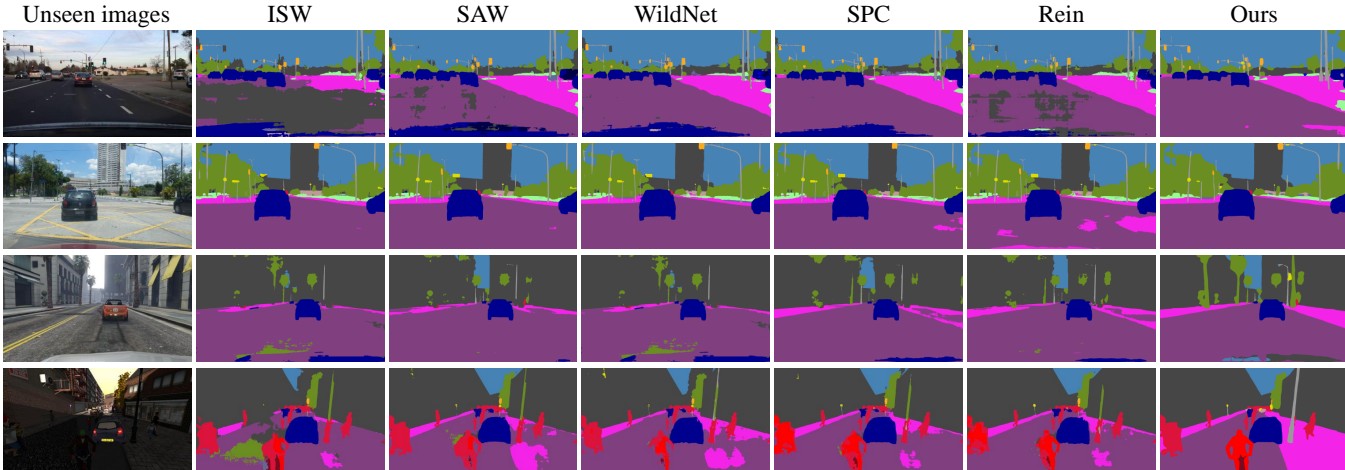

**Figure 5: Visual segmentation results on unseen target domains under the C → B, M, G, S setting. The proposed SET is compared with ISW [12], SAW [36], WildNet [26], SPC [21] and Rein [51].**

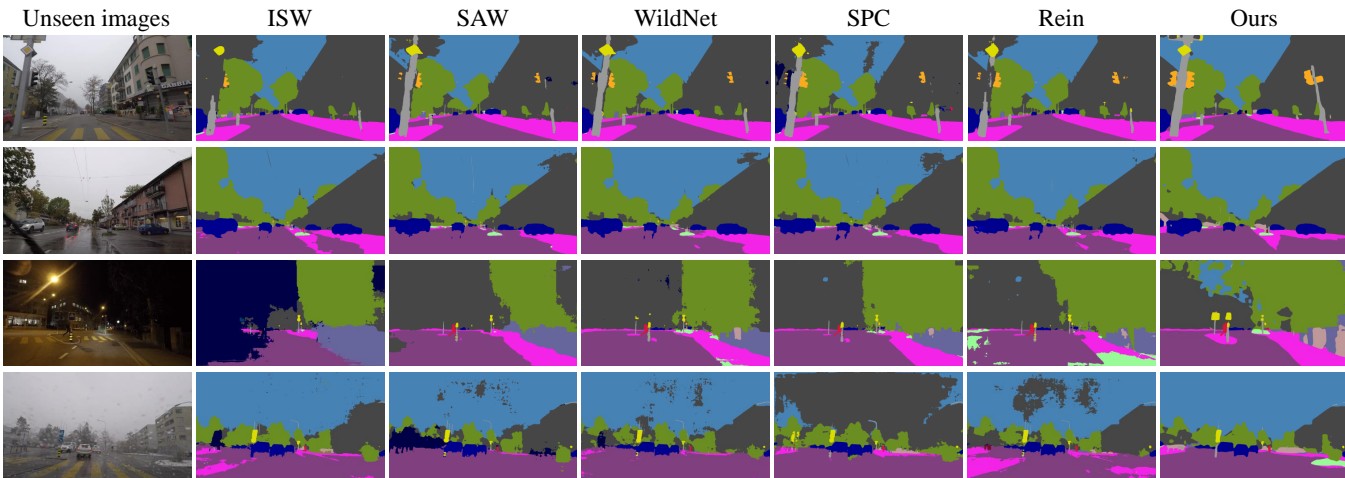

**Figure 6: Visual segmentation results on unseen target domains under the C → ACDC setting. The proposed SET is compared with ISW [12], SAW [36], WildNet [26], SPC [21] and Rein [51].**

domain. Table 7 compares the performance between SET and existing methods. It significantly outperforms existing ResNet based methods by at least 20% mIoU and Mask2Former based methods by at least 10% mIoU on all the adverse domains. Notably, compared with Rein [51] baseline, our SET shows improvements of 0.58%, 1.37%, 2.35% and 3.12% on mIoU in the fog, night, rain and snow domain, respectively.

## 4.5 Quantitative Segmentation Results

Fig. 5 demonstrates some visual segmentation results on unseen target domains under the C → B, M, G, S setting. Fig. 6 illustrates the visual segmentation results under the C → ACDC setting. On both settings, the segmentation results show that the proposed SET shows better pixel-wise prediction than the compared DGSS methods, especially in terms of the completeness of objects.

## 5 Conclusion

In this paper, we aim to fine-tune VFM for the down-stream task DGSS. While the VFMs have inherent generalization to out-of-distribution, how to exploit the style-invariant property of a VFM remains to be the bottleneck. We propose a Spectral-Decomposed Token (SET) learning scheme. As the content and style information in a scene reflects more from the low- and high- frequency components in the frequency space, we transform the frozen VFM features into the phase and amplitude components respectively. Then, spectral tokens are adapted to enhance the learning of task-specific knowledge within each branch. As the cross-domain differences mainly affects the amplitude branch, an attention optimization method is further proposed to mitigate the impacts of style variation. Extensive experiments under multiple cross-domain settings show the state-of-the-art performance of the proposed SET and its flexibility on a variety of VFMs.

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

Received 20 February 2007; revised 12 March 2009; accepted 5 June 2009
