# OpenReview forum: "Learning Spectral-decomposited Tokens for Domain Generalized Semantic Segmentation"
_acmmm.org/ACMMM/2024/Conference — MM2024 Poster_

### Official Review · Reviewer_cLdA · 2024-05-22

**Rating:** 5
**Confidence:** 2

**Summary:**

For the goal of DGSS, this paper presents a frequency-based fine-tuning method for the large CV models. By adding learnable amplitude and phase tokens, the proposed method achieve better performance than previous DGSS methods.

**Strengths:**

+ As illustrated in Fig.2, the motivation of this paper is clear.
+ The presentation of the proposed method is clear and good.
+ The idea to fine-tune features from large models from the perspective of spectrum is novel for the reviewer.
+ The experimental results show the superiority of the proposed method.

**Limitations:**

- This paper may need proof-reading. The "reference stage" in 3-rd line of introduction may be "inference stage". Fig.1 and Fig2 are not mentioned in the introduction.
- Why the attention optimization is not used in the phase branch?

**Suitability:**

2

---

### Official Review · Reviewer_e5jK · 2024-05-22

**Rating:** 3
**Confidence:** 3

**Summary:**

The paper proposes a novel Spectral-dEcomposed Token (SET) learning framework that uses spectral decomposition and learnable tokens to improve the domain generalization capabilities of Vision Foundation Models (VFMs) for semantic segmentation, achieving state-of-the-art performance across various unseen target domains.

**Strengths:**

The paper is well written, the overall structure is clear, and the experiments are comprehensive.

**Limitations:**

The core idea of this paper is to perform domain transformation on the side features of CNNs, followed by multiple feature fusion methods. This idea is not novel; I have seen it frequently in related tasks. I think the major flaw of this work is that the authors list many experiments to demonstrate that the method is good and better than its competitors, but they do not discuss and reveal why adopting this method can achieve such good results. I want the authors to explain this point very clearly in the rebuttal and include it in the final text. Since the idea is not novel, I hope the authors can delve deeper into the fundamental nature of the work.

Other flaws include the t-SNE visualization in Figure 4. The new cluster results seem much worse than the baseline.

**Suitability:**

3

---

### Official Review · Reviewer_MF6q · 2024-06-06

**Rating:** 4
**Confidence:** 3

**Summary:**

In this paper, the authors present a framework named SET for domain-generalized semantic segmentation. In this framework, the frozen VFM features are first decomposed into the phase and amplitude components in the frequency space and then refined by learned phase tokens and amplitude tokens, respectively. The SET gets SOTA in several datasets.

**Strengths:**

1. Based on VFM, SET achieves remarkable performance in domain-generalized semantic segmentation with very few learnable parameters.
2. The article presents exhaustive and convincing experiments, complemented by comprehensive and well-executed visualizations.

**Limitations:**

The efficacy of "Spectral Decomposition" in domain-generalized semantic segmentation has been validated in [A]; your fine-tuned network architecture resembles to [B], which also employs VFM for the same task. Your technical innovation lies in the "attention optimization." Thus, I view your work's contribution as a successful exploration of Spectral Decomposition in VFM-based domain-generalized semantic segmentation, accompanied by adaptive improvements. So my main concern lies in :
1. In-depth Analysis of Attention Optimization: Explain why this technique works in your framework, perhaps with examples or visualizations of the attention matrix M.
2. Practicality of SET: As I know, FFT and IFFT can be computationally intensive, please analyze the computational complexity of SET or provide inference time costs to gauge its practical feasibility.

Additionally, there are minor inaccuracies in your manuscript:

3. Figure 3: The figure should indicate that it's the phase (not phase token) that is added to β(ρk).
4. Shouldn't the dimensions of Xf be d×H×W instead of H×W?

[A] FDA: Fourier Domain Adaptation for Semantic Segmentation

[B] Stronger, Fewer, & Superior: Harnessing Vision Foundation Models for Domain Generalized Semantic Segmentation

**Suitability:**

2

---

### Official Review · Reviewer_LPvg · 2024-06-07

**Rating:** 5
**Confidence:** 4

**Summary:**

Effectively fine-tuning VFM for DGSS has recently become an open research topic for the vision community. In this paper, the authors present a novel Spectral-dEcomposed Token (SET) learning framework to advance the frontier. Delving into further than the existing fine-tuning token & frozen backbone paradigm, the proposed SET especially focuses on the way learning style-invariant features from
these learnable tokens.  After the decomposition, style variation primarily impacts the token-based feature enhancement within the amplitude branch. To address this issue, the authors further develop an attention optimization method to bridge the
gap between style-affected representation and static tokens during inference. Extensive cross-domain experiments under a variety of
backbones and VFMs show state-of-the-art performance.

**Strengths:**

--- The organization of this paper is clear and easy to follow.

--- The proposed Spectral-dEcomposed Token (SET) is novel and effective.

--- Extensive cross-domain experiments under a variety of backbones and VFMs show state-of-the-art performance.

**Limitations:**

--- In FSDR [A], the authors first converted the latent embedding into frequency space (64FCs) with Discrete Cosine Transform and found that only low-frequency and high-frequency components capture more domain variant features as compared with middle-frequency components. While in this manuscript, the authors used ONE global Amplitude. This needs to be explained in detail.

--- The article primarily addresses the factor of style. While it's important to acknowledge that style is a significant factor in domain drift, particularly in semantic segmentation tasks, it's also crucial to recognize a limitation in this approach: the predominant focus on style by many works, including this paper, limits a comprehensive understanding of the issue.

--- Lacks of some closely related work to be discussed[B-C].

[A] FSDR: Frequency Space Domain Randomization for Domain Generalization.
[B] Test-time Fourier Style Calibration for Domain Generalization.
[C] Adversarial Style Mining for One-Shot Unsupervised Domain Adaptation.

**Suitability:**

3

---

### Official Review · Reviewer_KyQs · 2024-06-07

**Rating:** 3
**Confidence:** 3

**Summary:**

The paper proposed a novel method based on spectral-decomposed for domain generalisation. It can be integrated into existing VFMs and bring improvement by comparing with different methods on many datasets.

**Strengths:**

1. Decomposed in frequency level is good for separating the structural and style components in the image features.
2. The proposed method has good generalisation ability among different datasets. The feature after processing is more uniform as seen in Figure 4, which removes the domain gap between different datasets.

**Limitations:**

1.
```
The phase component, which contains more low-frequency components on the scene content, is relatively stable despite the cross-style variation.
```
This claim is not always correct. Low-frequency components contain more structural information, which includes low-frequency and high-frequency information. Although low-frequency components represent mainly the large-scale structures in the image, high-frequency components also contain structures as well. In some cases, the phase may contain more high-frequency components than low-frequency components. The distribution of phase information is not directly related to the amount of the frequency component. In fact, since the style transfer doesn't change the structural information, the phase wouldn't have changed much in the first place.

2.
```
Figure 3:
```
The amplitude and phase are two interrelated components. But in the framework, they are processed independently, and they will be composed in the end, which raise my concern.

3. Lack of ablation study. The contribution of each section is not clear.

4.
```
Received 20 February 2007; revised 12 March 2009; accepted 5 June 2009
```
This sentence at the end of References does not make sense.

**Suitability:**

2

---

### Meta-Review · Area_Chair_N5SE · 2024-07-01

**Recommendation:** Accept (Poster)
**Confidence:** 5

**Metareview:**

This paper received three weak accept, one weak reject and one reject final ratings from the reviewers. The main concern of some reviewers is that this work lacks discussion and explanation about why the proposed approach works well for semantic segmentation. But the experimental results can sufficiently confirm the effectiveness of the proposed approach. AC agrees that this paper benefits from good writing and interesting idea. However, the authors are encouraged to make the necessary changes to the best of their ability.